# Association between the COVID-19 pandemic and mental health in very old people in Sweden

Fanny Jonsson[1], Birgitta Olofsson [2,3], Stefan Söderberg [4‡], Johan Niklasson [1‡]*

1 Community Medicine and Rehabilitation, Sunderby Research Unit, Umeå University, Umeå, Sweden,
2 Department of Nursing, Umeå University, Umeå, Sweden, 3 Department of Surgical and Perioperative Science, Orthopedics, Umeå University, Umeå, Sweden, 4 Department of Public Health and Clinical Medicine, Umeå University, Umeå, Sweden

‡ SS ans JN are contributed equally to this work as last authors.
* johan.niklasson@umu.se

**Data Availability Statement:** Data are available from the The Biobank Research Unit, Umeå university, for researchers who meet the criteria for access to confidential data and have approval from Ethics Committee. More information can be found

## Abstract

### Background

During the COVID-19 pandemic, Sweden implemented social distancing measures to reduce infection rates. However, the recommendation meant to protect individuals particularly at risk may have had negative consequences. The aim of this study was to investigate the impact of the COVID-19 pandemic on very old Swedish peoples' mental health and factors associated with a decline in mental health.

### Methods

We conducted a cross-sectional study among previous participants of the SilverMONICA (MONItoring of Trends and Determinants of CArdiovascular disease) study. Of 394 eligible participants, 257 (65.2%) agreed to participate. Of these, 250 individuals reported mental health impact from COVID-19. Structured telephone interviews were carried out during the spring of 2021. Data were analysed using the $\chi^2$ test, t-test, and binary logistic regression.

### Results

Of 250 individuals (mean age: 85.5 ± 3.3 years, 54.0% women), 75 (30.0%) reported a negative impact on mental health, while 175 (70.0%) reported either a positive impact (n = 4) or no impact at all (n = 171). In the binary logistic regression model, factors associated with a decline in mental health included loneliness (odds ratio [95% confidence interval]) (3.87 [1.83–8.17]) and difficulty adhering to social distancing recommendations (5.10 [1.92–13.53]). High morale was associated with positive or no impact on mental health (0.37 [0.17–0.82]).

here: https://www.umu.se/en/biobank-research-unit/research/access-to-samples-and-data/ Please contact the Biobank Research Unit before writing the application for ethical approval: asa.agren@umu.se.

**Funding:** The SilverMONICA study was funded by FORTE, the Swedish Research Council for health, working life and welfare (2016-01074); the County Councils in Norrbotten and Västerbotten (Visare Norr and ALF); the Borgerskapet in Umeå Research Foundation; the Swedish Dementia Association; the Ragnhild and Einar Lundström Memorial Fund; the Erik and Anne-Marie Detlof Research Foundation; the Swedish Society of Medicine; Thuréus; the Strategic Research Program in Care Sciences (SFO-V, Sweden); and the King Gustaf V and Queen Victoria's Foundation of Freemasons. The COVID-19 follow-up was supported by Umeå University. The funders had no role in the study design, data collection and analyses, decision on whether to publish or preparation of this manuscript.

**Competing interests:** The authors have declared that no competing interests exist.

## Conclusions

A high percentage of very old people reported a negative impact on mental health from the COVID-19 pandemic, primarily from loneliness and difficulty adhering to social distancing measures, while high morale seemed to be a protective factor.

## 1. Introduction

Since the beginning of the COVID-19 outbreak, there have been over 525 million confirmed cases and almost 6.3 million deaths globally [1]. In Sweden, a country with almost 10.5 million inhabitants [2], more than 2.5 million cases have been confirmed, and almost 19,000 deaths have been reported [1], with the highest death rate among the oldest old [3]. The World Health Organisation declared the COVID-19 outbreak a pandemic on March 11, 2020 [4], and soon after, many countries implemented various states of social distancing to reduce the spread of infection. In Sweden, people over 70 years were advised to limit social interactions to a higher degree than the rest of the population [5].

As older people were at higher risk of suffering negative outcomes related to COVID-19 regardless of comorbidities [6], social distancing was implemented to protect those particularly at risk. However, the recommendation thought to act as a protective measure has also negatively impacted mental health. Several studies have described mental health consequences from social distancing during COVID-19, such as depressive symptoms, anxiety, poor sleep quality [7], loneliness [8, 9] and subjective health impacts [10]. However, contradictory results have been presented, such as low levels of depression and anxiety among the participants [11].

Possible explanations for these positive outcomes include mental resilience through life experiences such as growing up during the Second World War years and exposure to previous pandemics [11], as well as inherent aspects of mental resilience. One of the latter is morale, defined by Lawton in the 1970s as "*a generalized feeling of well-being with diverse specific indicators such as freedom from distressing symptoms, satisfaction with self, feeling of syntony between self and environment, and ability to strive appropriately, while still accepting the inevitable*" [12]. Higher morale, that is, future-oriented optimism [13], is associated not only with increased longevity [14, 15], but with lower risk of depressive disorders as well [16].

However, the studies mentioned above evaluated older people's experiences only during the first months of social distancing during COVID-19, and none were conducted with samples that exclusively included the oldest old (aged 80 or older) [7–11]. Lastly, it is unclear whether these conclusions are applicable to a Swedish context since these studies were conducted in countries with preventive measures that led to society being shut down to a greater extent than it was under the Swedish COVID-19 strategy, which was based on voluntary action and personal responsibility [17].

The purpose of this study was to investigate the impact the COVID-19 pandemic had on Swedish older peoples' mental health and factors associated with a decline in mental health.

## 2. Material and methods

### 2.1 Study design

The SilverMONICA COVID-19 survey is a follow-up of individuals who previously participated in the SilverMONICA (MONItoring of Trends and Determinants of CArdiovascular disease) study, which was a follow-up study of previous participants from the Northern

Sweden MONICA study performed in 1999. Inclusion criteria were being 80 years or older during the study period 2016–2019 and living in either of the two northernmost counties of Sweden (Norrbotten or Västerbotten). The MONICA [18] and SilverMONICA [19] studies have previously been presented in detail.

## 2.2 Participants

The inclusion criteria for the SilverMONICA COVID-19 survey were previous participation in the SilverMONICA study when consent to future contacts was obtained. No exclusion criteria were implemented. A total of 394 individuals were eligible for participation, of whom 257 (65.2%) agreed to participate. Reasons for not participating were the following: emigrated or unable to reach (n = 11), declined participation (n = 57), declined because of health-related reasons and/or cognitive impairment (n = 58), or deceased (n = 11). Of the 257 individuals who participated, 250 answered the survey statement, "I feel that the COVID-19 pandemic has affected my mental health", and these individuals constituted the final sample.

## 2.3 Data collection

The Silver-MONICA study was performed with interviews in the participants' own homes, but due to the risk of spreading COVID-19, interviews, including home visits, were deemed inappropriate for the Silver MONICA COVID-19 follow-up. Instead, structured telephone interviews were carried out. Data collection occurred during the spring of 2021, starting in January and ending in June. An experienced research nurse conducted a vast majority of telephone interviews, while a few of the interviews were led by a specially trained student nurse, and both were purposely trained before the study. Participants were offered breaks during the interview process, as well as the possibility to continue later if needed. Most respondents could complete the interview independently, while in a few interviews were conducted with the support of relatives.

## 2.4 Measurements

**2.4.1 Outcome variable.** The impact on mental health was surveyed with the question, "I feel that the COVID-19 pandemic has affected my mental health", where answer alternatives included "for the better", "unaffected", and "for the worse" (0 = positive/no impact, 1 = negative impact).

**2.4.2 Sociodemographic factors.** For educational level, 13 or more years of schooling indicated a high educational level in accordance with Waller et al. [20]. For residential status, answer alternatives included "residing alone", "residing with significant other", and "residing with children or grandchildren" (0 = residing with other people, 1 = residing alone). Data regarding educational status, educational level, place of residence and residential status were retrieved from the SilverMONICA baseline (2016–2019).

**2.4.3 Health-related factors.** Self-rated health was based on the 36-Item Short Form Survey (SF-36) item "In general, would you say your health is:", and answer alternatives were dichotomized (0 = excellent/very good/good, 1 = fair/poor) [21, 22]. Self-rated health compared to one year ago was based on the SF-36 item "Compared to one year ago, how would you rate your health in general now?" [21, 22], and the answer alternatives were dichotomized (0 = better/about the same as one year ago, 1 = worse than one year ago).

Sense of security was dichotomised (0 = secure, 1 = insecure/hard to tell), where "hard to tell" was dichotomized as worse, with the justification that individuals who stated that it was hard to tell whether they felt secure or not could not, likely were insecure. Perceived loneliness

was dichotomised by merging the answers "rarely" and "never" and the answers "often" and "sometimes" (0 = no, 1 = yes).

The 15-item version of the Geriatric Depression Scale (GDS-15) [23, 24] is an assessment of depressive symptoms in older individuals. The scale consists of 15 questions, with scores ranging from 0 to 15. A score $\geq 5$ is considered to indicate depression, following Conradsson et al. [25], $\geq 14$ answered questions were regarded as a minimum for inclusion in the study.

The Philadelphia Geriatric Center Morale Scale (PGCMS) was used to assess morale in very old people [12]. The scale consists of 17 questions with answer alternatives "yes" or "no", where points are given for answers indicating high morale. Scores range from 0 to 17, through which level of morale is categorised as either low (0–9), mid-range (10–12), or high (13–17) [26]. The psychometric properties of the Swedish translation of the scale and its feasibility among very old people have been found satisfactory [27]. In line with a previous study by Niklasson et al. [16], $\geq 12$ answered questions were regarded as a minimum for inclusion in the present study.

The Mini-Mental State Examination (MMSE) [28] assesses cognitive function that includes orientation, registration, attention and calculation, recall, language, and copying. Scores range from 0 to 30, with a score of $\leq 23$ indicating significant cognitive impairment [29]. Dependence in instrumental activities of daily living (I-ADL) and personal activities of daily living (P-ADL) was measured with the Katz ADL staircase [30]. To be deemed as independent in I-ADL the participants had to be independent in all the following activities: cleaning, shopping, transport, and cooking. To be deemed independent in P-ADL, the participants had to be independent in all the following activities: bathing, dressing, toileting, transfer, continence, and eating. Mean gait speed at the usual pace (metres per second [m/s], measured over 2.4 m) was included as a surrogate marker for frailty by Weidung et al. [31]. Individuals were considered to have impaired hearing if the respondent reported that they could not hear someone speaking in a normal tone of voice from 1 metre away, with or without hearing aids. Data regarding MMSE score, dependence in I-ADL, dependence in P-ADL, gait speed and impaired hearing were retrieved from the SilverMONICA baseline (2016–2019).

Participants were asked if they knew if they had had COVID-19 infection. Approximately 15% of the participant's medical records were randomly selected to be reviewed for COVID-19 test results.

**2.4.4 Social factors.** Difficulty adhering to social distancing recommendations was based on the question "How easy has it been for you to follow the Public Health Agency of Sweden's recommendation regarding social distancing?". The variable was dichotomised by merging the answer alternatives "very easy" and "somewhat easy", and the answer alternatives "very difficult", "somewhat difficult" and "I do not know", with the justification that individuals who stated that they did not know whether it had been easy to adhere to social distancing recommendations cannot have found it easy to do so (0 = easy, 1 = difficult).

The answer alternatives were dichotomised for subjective impact on the social situation by COVID-19 (0 = positive/no impact, 1 = negative impact). Experience of not receiving the required amount of help was based on the question, "Do you receive the help that you feel you require from others (home help services, relatives etc.)?", and answers were dichotomized (0 = yes, 1 = no). For regular visits from home-help services, answer alternatives included "no" and "yes" (0 = no, 1 = yes), and "I do not know" answers were coded as missing.

The variables "frequency of physical contacts" and "frequency of non-physical contacts" were further divided into the frequency of physical and non-physical contacts with family members, relatives, friends, neighbours, and people not otherwise specified. For each subcategory, answer alternatives included "more contact", "less contact", "no difference in contact", "I have no contact with [subcategory]", and "I do not know". Subjects who selected the answer alternative "I do not know" were coded as missing.

 

For the variable "frequency of physical contacts", we sought to investigate any eventual decrease in frequency. Therefore, subcategories were dichotomised by merging the answer alternatives "more contact", "no difference in contact", and "I have no contact with [subcategory]", thus leaving dichotomous subcategories (0 = higher/same/no frequency of physical contacts, 1 = lower frequency of physical contacts).

In contrast, for the variable "frequency of non-physical contacts", we sought to investigate any eventual increase in frequency. Subcategories were therefore dichotomised by merging the answer alternatives "less contact", "no difference in contact", and "I have no contact with [subcategory]", thus leaving dichotomous subcategories (0 = lower/same/no frequency of non-physical contacts, 1 = higher frequency of non-physical contacts). The subcategory "one/more social networks" was created for both variables by computing each dichotomised subcategory.

Social distancing in this study was defined as avoiding crowded places, avoiding social contacts outside the family or persons in your household and keeping a longer physical distance from other people if crowded places had to be visited.

### 2.5 Statistics

For the univariable analyses, dichotomous variables were analysed using a $\chi^2$ test for independence or Fisher's exact test, while continuous variables were analysed using an independent-samples t-test. To compare those with and without mental impact from COVID-19 some questions with multiple answer alternatives were dichotomized. Binary logistic regression was performed to analyse the influence of predictor variables on the odds that respondents would report a negative impact on their mental health by the COVID-19 pandemic. Any variable with a univariate $p$-value of <0.15 was selected as a candidate for the multivariable analysis [32]. For competing variables in similar areas, variables were selected based on the knowledge and experience of the author group to limit the number of independent variables to approximately one per 10 participants with the investigated outcome (negative impact on mental health). The final binary logistic regression model contained 8 covariates (sex, educational level, self-rated health compared to one year ago, perceived loneliness, PGCMS score, dependence in P-ADL, subjective impact on social situation by COVID-19, and difficulty adhering to social distancing recommendations).

All statistical tests were two-tailed. $P$-values of <0.05 were considered statistically significant. All statistical analyses were conducted using SPSS, Version 27.0.1.0.

## 3. Results

### 3.1 Sample characteristics

In the COVID-19 follow-up study population (n = 250), 135 (54.0%) were female. The mean age of the participants was 85.5 ± 3.3 years (range 81–96) at the time of the COVID-19 follow-up survey. Of the 250 respondents, 75 (30.0%) considered their mental health to be negatively affected by the COVID-19 pandemic, while 175 (70.0%) experienced either a positive impact (n = 4) or no effect at all (n = 171).

### 3.2 Sociodemographic factors

Sociodemographic factors concerning mental health are listed in Table 1, and these data were retrieved from the original SilverMONICA study. There were no statistical differences between those with and those without impact on mental health from COVID-19 regarding age, sex, mean number of years in school, level of education, place of residence, or residential status.

**Table 1. Association between sociodemographic factors and mental health during the COVID-19 pandemic (N = 250).**

|  | Negative impact on mental health by COVID-19 | | |
|---|---|---|---|
|  | Yes (N = 75) | No (N = 175) |  |
|  | n (M ± SD) (%) | n (M ± SD) (%) | *p* |
| Age (years) | 75 (85.4 ± 3.0) | 175 (85.5 ± 3.5) | 0.869 |
| Sex (female) | 48 of 75 (64.0) | 87 of 175 (49.7) | 0.053 |
| Educational status (years of schooling)* (N = 247) | 74 (10.5 ± 3.8) | 173 (10.2 ± 4.0) | 0.535 |
| Educational level (high)* (N = 247) | 24 of 74 (32.4) | 38 of 173 (22.0) | 0.115 |
| Place of residence (number of inhabitants)* (N = 239) |  |  | 0.733 |
| • Urban (≥15,000 inhabitants)* | 37 of 72 (51.4) | 79 of 167 (47.3) |  |
| • Semiurban (1,000–14,999 inhabitants)* | 19 of 72 (26.4) | 43 of 167 (25.7) |  |
| • Rural (<1,000 inhabitants)* | 16 of 72 (22.2) | 45 of 167 (26.9) |  |
| Residential status (residing alone)* (N = 244) | 38 of 75 (50.7) | 79 of 169 (46.7) | 0.670 |

* Data retrieved from the SilverMONICA study (2016–2019).

Percentages are reported for dichotomous variables, while continuous variables are presented as mean ± standard deviation. Dichotomous variables were analysed using the $\chi^2$ test for independence, while continuous variables were analysed using an independent-samples t-test.

High level of education: ≥13 years of schooling.

## 3.3 Health-related factors

Health-related factors associated with the negative impact on mental health by COVID-19 were self-rated health, self-rated health worse than a year ago, perceived loneliness, GDS-15 score ≥5, and PGCMS ≥13 (Table 2). Table 2 also presents data from the original SilverMO-NICA study: MMSE-score, I-ADL, P-ADL, Gait speed and Hearing.

**Table 2. Association between health-related factors and mental health during the COVID-19 pandemic (N = 250).**

|  | Negative impact on mental health by COVID-19 | | |
|---|---|---|---|
|  | Yes (N = 75) | No (N = 175) |  |
|  | n (M ± SD) (%) | n (M ± SD) (%) | *p* |
| Self-rated health (poor) (N = 247) | 17 of 72 (23.6) | 20 of 175 (11.4) | **0.025** |
| Self-rated health compared to one year ago (worse) (N = 246) | 39 of 71 (54.9) | 46 of 175 (26.3) | **<0.001** |
| Sense of security (insecure) (N = 246) | 17 of 73 (23.3) | 25 of 173 (14.5) | 0.134 |
| Perceived loneliness (N = 241) | 47 of 71 (66.2) | 52 of 170 (30.6) | **<0.001** |
| GDS-15 score ≥5 (indicating depression) (N = 187) | 15 of 59 (25.4) | 2 of 128 (1.6) | **<0.001** |
| PGCMS score ≥13 (indicating high morale) (N = 221) | 18 of 68 (26.5) | 91 of 153 (59.5) | **<0.001** |
| MMSE score* (N = 243) | 72 (26.5 ± 3.8) | 171 (26.4 ± 3.5) | 0.900 |
| I-ADL (dependent)* (N = 247) | 36 of 75 (48.0) | 87 of 172 (50.6) | 0.814 |
| P-ADL (dependent)* (N = 246) | 7 of 75 (9.3) | 7 of 171 (4.1) | 0.134[FE] |
| Gait speed (m/s)* (N = 240) | 73 (0.7 ± 1.9) | 167 (0.7 ± 1.3) | 0.819 |
| Hearing (impaired)* (N = 241) | 4 of 73 (5.5) | 10 of 171 (5.8) | 1.000[FE] |

* Data retrieved from the SilverMONICA study (2016–2019).

Percentages are reported for dichotomous variables, while continuous variables are presented as mean ± standard deviation. Bold type indicates significant values ($p < 0.05$). Dichotomous variables were analysed using the $\chi^2$ test for independence (with Yates' continuity correction), while continuous variables were analysed using an independent-samples t-test.

*FE* Fisher's exact test was applied in place of the $\chi^2$ test for independence if >20% of cells had an expected cell count <5, and no expected cell count <1.

GDS-15: Geriatric Depression Scale 15-item version; PGCMS: Philadelphia Geriatric Center Morale Scale; MMSE: Mini-Mental State Examination; I-ADL: instrumental activities of daily living; P-ADL: personal activities of daily living.

Only 4 out of 77 participants in the COVID-19 follow-up sample stated that they had had the COVID-19 infection. Due to the high number of participants not answering whether they had had COVID-19, approximately 15% of the participant's medical records were reviewed concerning positive COVID-19 test results. Of 37 participants with their medical records reviewed, only one had had a positive COVID-19 test.

### 3.4 Social factors

Social factors associated with the negative impact on mental health by COVID-19 were difficulty following social distancing recommendations, negative subjective impact on social situation by COVID-19, and reduced number of physical contacts (Table 3). The participants reduced their physical contacts with all parts of their social network (family, relatives, friends, neighbours, and others). There was a substantial reduction in reported contacts with other groups of people, ranging from 42.4% up to 84.1%. Those with negative impact on mental health by COVID-19 reduced their contacts more, but not significantly more than those without a negative impact. Only when combining one or more of the parts of their social network there was a significant difference (p-value 0.034). Further, there was an larger increase in non-physical contacts among those with negative impact by COVID-19 compare to those without, however it was not significantly larger.

### 3.5 Predictors of decline in mental health

The strongest predictor for reported decline in mental health by COVID-19 was difficulty adhering to social distancing recommendations (Table 4). Other predictors were perceived

**Table 3. Association between social factors and mental health during the COVID-19 pandemic (N = 250).**

| | Negative impact on mental health by COVID-19 | | |
|---|---|---|---|
| | Yes (N = 75) | No (N = 175) | |
| | n (%) | n (%) | $p$ |
| Difficulty adhering to social distancing recommendations (N = 244) | 23 of 73 (31.5) | 12 of 171 (7.0) | **<0.001** |
| Subjective impact on social situation by COVID-19 (negative) | 68 of 75 (90.7) | 116 of 173 (67.1) | **<0.001** |
| Experience of not receiving the required amount of help (N = 185) | 3 of 57 (5.3) | 6 of 128 (4.7) | 1.000[FE] |
| Regular visits from home help services (N = 248) | 18 of 74 (24.3) | 39 of 174 (22.4) | 0.871 |
| Frequency of physical contacts (lower) | | | |
| • One or more social networks (N = 221) | 68 of 68 (100.0) | 143 of 153 (93.5) | **0.034**[FE] |
| • Family members (N = 248) | 61 of 75 (81.3) | 122 of 173 (70.5) | 0.105 |
| • Relatives (N = 241) | 61 of 74 (82.4) | 116 of 167 (69.5) | 0.052 |
| • Friends (N = 242) | 59 of 74 (79.7) | 121 of 168 (72.0) | 0.269 |
| • Neighbours (N = 243) | 41 of 73 (56.2) | 72 of 170 (42.4) | 0.066 |
| • People not otherwise specified (N = 231) | 58 of 69 (84.1) | 123 of 162 (75.9) | 0.230 |
| Frequency of non-physical contacts (higher) | | | |
| • One or more social networks (N = 206) | 44 of 63 (69.8) | 85 of 143 (59.4) | 0.206 |
| • Family members (N = 239) | 43 of 72 (59.7) | 83 of 167 (49.7) | 0.200 |
| • Relatives (N = 239) | 34 of 74 (45.9) | 58 of 165 (35.2) | 0.149 |
| • Friends (N = 235) | 32 of 72 (44.4) | 58 of 163 (35.6) | 0.253 |
| • Neighbours (N = 233) | 8 of 72 (11.1) | 18 of 161 (11.2) | 1.000 |
| • People not otherwise specified (N = 212) | 11 of 65 (16.9) | 13 of 147(8.8) | 0.140 |

Percentages are reported for dichotomous variables. Bold type indicates significant values ($p < 0.05$). Dichotomous variables were analysed using the $\chi^2$ test for independence.

*FE* Fisher's exact test was applied in place of the $\chi^2$ test for independence if >20% of cells had an expected cell count <5, and no expected cell count <1.

**Table 4. Logistic regression analysing factors associated with a decline in mental health during the COVID-19 pandemic.**

| | | | 95% CI for OR | |
| --- | --- | --- | --- | --- |
| | *p* | OR | Lower | Upper |
| *Sociodemographic factors* | | | | |
| Sex (female) | 0.797 | 1.10 | 0.53 | 2.28 |
| Educational level (high)* | **0.037** | 2.32 | 1.05 | 5.12 |
| *Health-related factors* | | | | |
| Self-rated health compared to one year ago (worse) | **0.045** | 2.17 | 1.02 | 4.62 |
| Perceived loneliness | **<0.001** | 3.87 | 1.83 | 8.17 |
| PGCMS score ≥13 (indicating high morale) | **0.014** | 0.37 | 0.17 | 0.82 |
| P-ADL (dependent)* | 0.736 | 1.30 | 0.29 | 5.91 |
| *Social factors* | | | | |
| Subjective impact on social situation by COVID-19 (negative) | **0.012** | 3.74 | 1.34 | 10.48 |
| Difficulty adhering to social distancing recommendations | **0.001** | 5.10 | 1.92 | 13.53 |

\* Data retrieved from the SilverMONICA study (2016–2019).

Model performance in test sample (N = 226)

Hosmer and Lemeshow goodness-of-fit test: $\chi^2$ = 3.810, df = 8, *p* = 0.874 (indicating support for the model)

Nagelkerke $R^2$ (pseudo-$R^2$): 0.405

Bold type indicates significant values (*p* < 0.05). OR: odds ratio; CI: confidence interval; COVID-19: coronavirus disease 2019; P-ADL: personal activities of daily living; df: degrees of freedom.

loneliness, subjective impact on the social situation by COVID-19, high educational level, and self-rated health compared with one year ago. High morale, according to the PGCMS score, was associated with less decline in mental health. Sex and dependence in P-ADL did not associate with a decline in mental health.

## 4. Discussion

This study shows that 30% of the very old participants experienced that the COVID-19 pandemic had negatively affected their mental health. Both social and health-related factors were independently associated with a decline in mental health.

The explanation for why 30% of study participants reported a negative impact on mental health by the COVID-19 pandemic is most likely multifactorial, including intra-individual, inter-individual, societal, and cultural aspects. The seemingly low frequency of COVID-19 infections in the sample suggests that the negative impact on mental health is mainly an indirect effect, such as social distancing, and not directly caused by COVID-19 itself or the post-COVID syndrome.

The three factors with the highest odds ratio of negative impact on mental health by the COVID-19 pandemic in our final logistic regression model were all related to social factors. The three factors were: Difficulty adhering to social distancing recommendations, perceived loneliness and feeling that the social situation was negatively impacted by the COVID-19 pandemic. They will be further discussed below.

First, when asked whether the participants had found it difficult to follow restrictive measures recommended by the Public Health Agency, respondents who reported difficulty adhering to these recommendations had a higher risk for a decline in mental health in our logistic regression model. The multifaceted nature of this finding should be considered, encompassing cognitive (i.e., difficulty accessing and understanding recommendations) and social (i.e., difficulty being away from family and loved ones) dimensions. In a previous study by Gustavsson

et al. [17], 88.5% of participants found recommendations from government authorities in Sweden to be clear and concise, thus supporting the particular importance of the social aspect of the question (i.e., difficulty being away from loved ones). Mean MMSE scores and educational status did not differ between our groups, supporting the lesser importance of cognitive aspects. Conclusions should be made with caution, though, as the MMSE scores are from the Silver-MONICA baseline and, therefore, a few years old. There are many possible reasons why some individuals could find it difficult to follow social distancing recommendations. One feasible explanatory model is inadequate coping strategies when facing difficulties. Previous studies have found that adaptive coping strategies such as positive thinking, active stress coping, and social support are significant predictors of better mental health in a pandemic setting [33, 34], thus supporting the promotion of such adaptive coping strategies as a preventive measure during future pandemics. Worth noting is another study that after the Swedish Public Health Agency issued the non-mandatory recommendations to avoid crowded places found a sharp drop in visits to overcrowded places for 70-year-olds, which also resulted in a decline in severe COVID-19 cases for the same age-group [35].

Second, perceived loneliness was associated with a negative impact on mental health. This association between perceived loneliness and mental health is consistent with international results during the beginning of the COVID-19 pandemic [8, 36–38]. Interestingly, both those with negative impact on mental health and those without reduced their physical contact. Further, no significant association was found between residential status (living alone versus living with someone) and impact on mental health. Developing effective interventions for loneliness has proven more difficult. Though there might be no simple solution, researchers have found that engagement of older adults in social groups and communities reduces loneliness and its adverse effects [39–41] which is particularly difficult to implement during a pandemic.

Third, negative impact on their social situation was also reported significantly more among those with a negative impact on mental health from COVID-19 compared to those without. Similar results have been found by other authors [42, 43]. To further explore this we asked the participants about changes in frequency of their contacts with other people and there were reduction in physical contacts and an increase in non-physical contacts but there were hardly any differences between those who reported a negative impact on mental health by COVID-19 compared to those without a negative impact. This could be due to the sample size, how the questions were stated or that there were no difference. Finally, worth mentioning are the four participants who indicated a positive impact by COVID-19 pandemic on their mental health. This small group is interesting, however too small for statistical testing and in all our statistical calculations they were therefore combined with those who reported no impact on mental health by COVID-19. We can only speculate on the reasons for the positive impact on mental health from social distancing measures during COVID-19, but perhaps the participant had unwanted social contacts before COVID-19 that they now had an excuse to withdraw from.

Another important factor for mental health during COVID-19 pandemic could be health aspects, particularly psychiatric health issues. The prevalence of GDS-15 scores indicative of depression was only 9.1% in the total sample. This is lower than previous results, with studies reporting prevalence of depression during the COVID-19 pandemic ranging from 22.2% to 46.4% [17, 44, 45]. This could partly be explained by differences in data collection, measuring scales, and age of participants and may limit the comparability of study results. However, among respondents reporting a negative impact on mental health by the COVID-19 pandemic, 25.4% had GDS-15 scores indicative of depression.

A previous study by Ausín et al. suggested that women experienced more depression, anxiety, and loneliness during the COVID-19 pandemic [46]. However, the present study found no significant association between women and negative mental health impacts. Possible

explanations may be our selection of participants in the study, lack of power or because the study only examined very old individuals.

High morale was associated with a positive or no impact on mental health. There may be various reasons for this; one possible explanation is that high morale is a mental resilience factor, as seen by other authors with other mental resilience factors [47, 48]. Our findings indicate that promoting morale may be beneficial during future pandemics.

Surprisingly, high educational level was associated with a decline in mental health in the binary logistic regression model. Explanations can only be speculative: it is possible that these individuals are more dependent on social interactions, have a higher demand for cultural events that were withheld from them, or that they are more susceptible to negative mental health impacts from pandemics. The opposite result was seen in a younger sample by Creese et al. [8].

Since this study showed that a relatively high percentage of older adults had a reduction in mental health due to social distancing, the implication of this study is that efforts must be taken in future pandemics to identify individuals at risk and reduce their suffering. A program to reduce the mental consequences probably needs to be multifactorial, where this study identified at least some important health-related and social factors.

## 4.6 Strengths and limitations

A particular strength of this study is that the participants were very old: all participants were over 80. Even though this specific age group is at higher risk of contracting the serious disease and subsequently the most targeted by distancing measures, the coverage of very old individuals has been limited in previous studies. Structured telephone interviews, with the possibility of requesting and providing clarification, provided additional information not necessarily available through questionnaires or registry-based studies. The availability of SilverMONICA baseline data allowed the use of complementary data that were not accessible through telephone interviews. Another strength of this study was the high participation rate (65.2%).

However, a few points need to be considered when interpreting our results. First, the question used to assess the possible impact on mental health by the COVID-19 pandemic has not been previously validated. Using a one-question indicator to assess mental state is common and can be effective [49]; however, a set of questions would probably better capture the essence and enable a more fine-tuned cut-off for a decline in mental health. Still, a few factors support the claim of a decline in mental health, such as the changes in self-rated health and sense of security as well as the higher prevalence of depressive symptoms and perceived loneliness among respondents stating a negative impact on mental health by COVID-19. Further, dichotomizing questions with multiple answer alternatives, might lead to the loss of some information and precision [50].

There are some possible biases in this study. One is volunteer bias since only participants were willing to participate in MONICA, SilverMONICA, and this follow-up survey constituted the final research sample. When studying individuals 80 years old or older, survival bias also needs to be acknowledged. Finally, the subject's ability to participate in this follow-up study was influenced by hearing capability as well as cognitive ability, and this selected the healthier individuals.

The study's cross-sectional design limits claims regarding causality. Some data were collected in 2016–2019 (the SilverMONICA baseline); hence, these data should be interpreted with reservations, given possible changes since then.

Another aspect to consider is the changes in the epidemic situation during data collection. At the beginning of the study, Sweden was in the middle of its second wave of COVID-19.

Infection rates were high, especially among individuals living in retirement homes or receiving home help services [51]. Vaccination against COVID-19 had been initiated, with residents of long-term care facilities, healthcare workers, and older adults being prioritised [52]. In contrast, at the end of data collection, Sweden was at the end of its second wave of COVID-19, infection rates were decreasing, and many study subjects had already received their first vaccination.

## 4.7 Conclusion

A relatively high percentage of very old people had a negative impact on mental health from social distancing related to the COVID-19 pandemic. The cause is probably multifactorial, and this study showed that factors such as perceived loneliness and difficulty adhering to social distancing measures are important. Still, also resilience factors such as high morale seemed to be protective. These factors are important if future pandemics' negative impact on mental health for older people from future pandemics should be prevented.

## Acknowledgments

We are thankful to all participants in the SilverMONICA study, who devoted many hours to the baseline examination and this follow-up. We thank Ronja Messmer and Anna Olofsson for performing the interviews. Anna Olofsson, together with Mirjam Söderberg prepared the dataset, and Robert Lundqvist contributed statistical analyses. Lastly, we thank all the researchers and research assistants involved in SilverMONICA and the SilverMONICA follow-up survey, as well as the Biobank Research Unit at Umeå University.

## Author Contributions

**Conceptualization:** Fanny Jonsson, Birgitta Olofsson, Stefan Söderberg, Johan Niklasson.

**Data curation:** Fanny Jonsson, Birgitta Olofsson, Stefan Söderberg, Johan Niklasson.

**Formal analysis:** Fanny Jonsson, Birgitta Olofsson, Stefan Söderberg, Johan Niklasson.

**Funding acquisition:** Birgitta Olofsson, Stefan Söderberg, Johan Niklasson.

**Investigation:** Birgitta Olofsson, Stefan Söderberg, Johan Niklasson.

**Methodology:** Fanny Jonsson, Birgitta Olofsson, Stefan Söderberg, Johan Niklasson.

**Project administration:** Birgitta Olofsson, Stefan Söderberg, Johan Niklasson.

**Supervision:** Birgitta Olofsson, Stefan Söderberg, Johan Niklasson.

**Writing – original draft:** Fanny Jonsson, Birgitta Olofsson, Stefan Söderberg, Johan Niklasson.

**Writing – review & editing:** Fanny Jonsson, Birgitta Olofsson, Stefan Söderberg, Johan Niklasson.

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
