## [Decision Letter · Decision Letter 0]

25 Jul 2023

PONE-D-22-23481Home alone: association between the COVID-19 pandemic and mental health in very old peoplePLOS ONE

Dear Dr. Niklasson,

Thank you for submitting your manuscript to PLOS ONE. After careful consideration, we feel that it has merit but does not fully meet PLOS ONE’s publication criteria as it currently stands. Therefore, we invite you to submit a revised version of the manuscript that addresses the points raised during the review process.

We look forward to receiving your revised manuscript.

Kind regards,

Wenjing Zhao, Ph.D.

Academic Editor

PLOS ONE

Journal Requirements:

Additional Editor Comments (if provided):

In light of both the reviewers comments please submit the revised manuscript.

Reviewers' comments:

Reviewer's Responses to Questions

**Comments to the Author**

1. Is the manuscript technically sound, and do the data support the conclusions?

Reviewer #1: Yes

Reviewer #2: Partly

2. Has the statistical analysis been performed appropriately and rigorously? 

Reviewer #1: Yes

Reviewer #2: No

3. Have the authors made all data underlying the findings in their manuscript fully available?

Reviewer #1: Yes

Reviewer #2: No

4. Is the manuscript presented in an intelligible fashion and written in standard English?

Reviewer #1: Yes

Reviewer #2: No

5. Review Comments to the Author

Reviewer #1: This manuscript is significant and relevant to the current post-Covid scenario, where it is trying to find the relationship between mental health and social distancing, also relate it to multivariable like sex, education level, and place of residence. The study is mainly about the oldest old population who were more severely, and in Sweden, the COVID-19 social distancing norms were primarily focused on the aged people (above 70 years). The author analyzed that only 30% support a negative impact on mental health. In contrast, a larger group of around 70% supports no or positive effects on mental health and relates it with high morale. This point need a more clarification.

The Title of the manuscript is concise and gives a clear idea about the whole manuscript.

The abstract summarizes well with well-defined sections like background, methods, results, and conclusion. There is enough information to make readers understand immediately, though here, the author might have a check on the repetition of data (30%) in results and conclusion.

The introduction gives a brief background and contemporary scenario related to this study. Appreciate the data retrieved from the World Health Organization about the number of confirmed COVID cases and death; the author might add the date when it has been retrieved. This study has been related to the survey and has appropriate research objectives, which can be more elaborate.

The method section is clear, well describes sources, ethical consent, interview details, and sample size. The author could clarify the use of different scales to avoid confusion. Here in part 2.5.4, the author well classifies the variable into physical contacts and non-physical contacts. However, the word social distancing could be defined briefly to avoid confusion.

The result presented goes the same with the purpose and method of the study. The tables added were clear. The source of data written by the author is Silver MONICA (2016-19); there needs to be clarity on whether the data source is the mentioned one or the follow-up study Silver MONICA COVID-19.

The discussion sections were the findings were well supported by the relevant work done with COVID-19 and mental health. The author aims to relate the result; however, the work is more specifically supported based on European countries. The author might refer to https://doi.org/10.1093/eurpub/ckac101, which analyzes the Swedish government's positive impact of age-specific non-mandatory norms. And there is an alternate use of females and women in the seventh paragraph of this section if the author could clarify the reason behind this.

Acknowledging the effort for mentioning the strength and limitations, which is very realistic, like biases and self-reported data. The author needs to include the implication of this study.

However, the conclusion needs a significant revision. It should be expanded and elaborative with the study's findings and future scope.

Reviewer #2: “Home alone: association between the COVID-19 pandemic and mental health in very old people”

CONTENT OF THE REVIEW

Identifying features

Title of chapter and authors: Home alone: association between the COVID-19 pandemic and mental health in very old people

General observation:

Manuscript is partially sufficient to look over the research objective. Even study also fail to capture previous research related to mental health and home alone. Discussion part also suffer with contemporary discourses and related facts.

Statistical part is partially fulfilling the scientific analysis criteria. Association between variable and data size in row and column is not statistically corrected. It is applicable for table 1, table 2, and table 3. Binary logistic method also needs to be crosschecked. Finding in table 4 is doubtful. please check again and see the value of odd ratio and corresponding interval value.

Data is not available publicly, so it is matter of ethical approval from concern body. The author has ethical approval for data source.

Manuscript has gone through various incorrect sentences and wrong sentence frame; Manuscript needs a lot of rigorous grammatical check.

Abstract

The paper has a lot of problems with grammar that start right at the beginning. See “Of 394 eligible participants, 257 (65.2%) agreed to participate. Of these, 250 32 individuals reported mental health impact from COVID-19 and constituted the final sample”.

In result section under abstract, it is not good to write detailed finding of odds ratio.

Please revise the text with better sentence structure and clarity and make it concise.

Introduction

The current text exhibits a deficiency in connectivity and coherence both within individual sections and across different sections. There is a need to establish a connection between the two units. The authors should incorporate relevant scholarly sources from previous studies. The introduction fails to effectively convey the essence of the title. The topic of mental health is noticeably lacking in the section on being home alone. The manuscript, to some extent, did not successfully convey the essence of the concept of being home alone. The precise definition of study is not provided. Similar to this, literature on mental health has failed to show any correlation with numerous cofactors from earlier research conducted within a country or around the world.

Data and Methodology

The manuscript tries to include information from both the SilverMONICA study and the follow-up study. The authors missed the opportunity to do the appropriate amount of analyzing the data. 250 people filled out the survey (line 103), but the numbers in Table 1, Table 2, and Table 3 do not match up in the rows and columns where they should. If the row data is 247, how is it split between columns 75 and 175? And it is also applicable for other numbers in row.

The subjective response is grouped together under the health-related factor. Does the author have any studies that show how to combine subjective responses with five groups into two? Please provide appropriate references. Similarly, “Self-rated health compared to one year ago was based on the SF-36 item “compared to one year 138 ago, how would you rate your health in general now” and responses are 0 = secure, 1 = insecure/hard to tell. Do author has any reference to club the category (insecure and hard to tell). Do author has any reference to club the category (insecure and hard to tell). Again, in reference to Geriatric Depression Scale, Philadelphia Geriatric Center Morale Scale Mini-Mental State Examination, how author decided the limit of score for particular case or event. Author does not provide any depth information related to score construction and related questions. Also, the variable under "social factor" is not enough to explain why the categories of responses might be merged.

It is also good to capture only relevant scale or score instead of multiple scale or score, and before that detailed and proper justification need to be address.

Results

Table 1 consist of association of sample data i.e., 250 and they have answered the related question. This sample is again classified in Yes (N = 75) and No (N = 175). But when it associated with educational status (years of schooling) * (N = 247), Educational level (high)* (N=247), Place of residence (number of inhabitants) * (N = 239), and Residential status (residing alone) * (N = 244), these sample is not matched with column sample. It is again applicable for table 2 and table 3. It requires further analysis of data. In table 4 logistic regression was perform for various Sociodemographic factors, Health-related factors, and social factors. It seems that data analysis does not perform well. Odd ratio with lower and upper limit explains different story. I recommend for further analysis. The author again highlighted various score and scale for mental health, but manuscript does not consider such information and calculating method of that score. GDS-15 score and MMSE score do not capture in multivariate analysis. Author should provide an analysis or information-why they are excluded from the analysis how it can be identified without any findings (Any variable with a univariate p-value of <0.15 was selected as a candidate for the multivariable analysis. LINE:213)

Majorly all the tables have numerous findings, but author should interpret it under different section. It is missing in manuscript.

Why chi2 test and FE fisher test used together? it is good to relay on specific method for all the variables.

Discussion

In this section author used “very old”. Is it previously described in manuscript anywhere. Please refer it. Is 30 percent or any number is directly associated with table 1. line 309 to 313 is not evidence-based statement. Kindly refer it with previous research. Please explain line 325 with various associated factor comes under GDS-15. The line “This could partly be explained by differences in data collection, measuring scales, and age of participants and may limit the comparability of study results. However, amongst respondents reporting a negative impact on mental health by the 330 COVID-19 pandemics,” could be the part of limitation of study

Logistic regression model seems incorrectly estimated please check again. Even in discussion section odds value should be the part of discussion but it is missed majorly. Please refer standard form of odds ratio, p value, lower limit and upper limit during framing the statement.

Overall, this manuscript needs a lot of regress work under following section-

• Previous research and key definition of objective term

• Grammatical error and sentence framing

• Data cleaning and required variable.

• Data analysis and interpretation of findings

• Discussion and logical statement.

6. PLOS authors have the option to publish the peer review history of their article (what does this mean?). If published, this will include your full peer review and any attached files.

Reviewer #1: No

Reviewer #2: No

---

## [Author Response · Author response to Decision Letter 0]

25 Aug 2023

Response to Editor and Reviewers

We thank the reviewers for their time and effort in reviewing our manuscript. Please find our response below, where reviewers’ comments have been slightly shortened.

Editor

1. The style has been updated to better adhere to PLOS ONE style requirements. 

2. In the ethics section, “Signed consent” has now been changed to “Written consent”.

3. Funding section has been corrected. 

Reviewer #1

1. The author analyzed that only 30% support a negative impact on mental health. In contrast, a larger group of around 70% supports no or positive effects on mental health and relates it with high morale. This point need a more clarification.

Thank you for this; however, we are a little bit uncertain how to clarify this. We hope the newly added implication and conclusion give the clarification needed.

2. The abstract gives enough information to make readers understand immediately, though here, the author might have a check on the repetition of data (30%) in results and conclusion.

We appreciate that you pointed out this repetition. We have now changed the conclusion in the abstract to (line 51-53)

“A high percentage of very old people reported a negative impact on mental health from the COVID-19 pandemic, primarily from loneliness and difficulty adhering to social distancing measures, while high morale seemed to be a protective factor.

3. The introduction data retrieved from the World Health Organization about the number of confirmed COVID cases and death; the author might add the date when it has been retrieved. 

Thank you for noticing. For unknown reasons, this information disappeared. The access date is now visible in reference number one (cited 2022-05-30), line 548.

4. Research objectives can be more elaborate. 

Thank you for pointing out this. The research question is now extended: 

“The purpose of this study was to investigate the impact the COVID-19 pandemic had on older peoples’ mental health and factors associated with a decline in mental health.”, line 103-104.

5. In the method section, the author could clarify the use of different scales to avoid confusion. 

We fail to understand the reviewer’s concern. We have reported all scales used in our study. One for assessing depressive symptoms (GDS-15), one for morale (PGCMS), one for cognition (MMSE) and so on. 

6. The word social distancing could be defined briefly to avoid confusion.

A definition of social distancing has now been added: 

“Social distancing in this study was defined as avoiding crowded places, avoiding social contacts outside the family or persons in your household and keeping a longer physical distance from other people if crowded places had to be visited.”, line 274-276.

7. There needs to be clarity on whether the data source is the mentioned one or the follow-up study Silver MONICA COVID-19 in the Results section.

Thank you for pointing out this. We have added words to clarify where the data came from, please see the revised manuscript, especially Result sections 3.1 to 3.3. 

Some data from the original SilverMONICA study (2016-2019) we believed was unproblematic, for instance, years in school. Other data might have changed, for example, MMSE and ADL-status, but we thought it provided some information rather than not using them. New data was difficult to get over the telephone during the pandemic.

8. The author might refer to https://doi.org/10.1093/eurpub/ckac101, which analyzes the Swedish government's positive impact of age-specific non-mandatory norms. 

Thank you for the recommendation. The reference was added in the Discussion section, line 408-415 and 678-680.

9. And there is an alternate use of females and women in the seventh paragraph of this section if the author could clarify the reason behind this.

Thank you for pointing this out. We have changed the wording to: 

“A previous study by Ausín et al. suggested that women experienced more depression, anxiety, and loneliness during the COVID-19 pandemic (43). However, the present study found no significant association between women and negative mental health impacts”, line 425-426.

10. The author needs to include the implication of this study.

We agree and have added the following paragraph at the end of the Discussion: “Since this study showed that a relatively high percentage of older adults had a reduction in mental health due to social distancing, the implication of this study is that efforts must be taken in future pandemics to identify individuals at risk and reduce their suffering. A program to reduce the mental consequences probably needs to be multifactorial, where this study identified at least some important health-related and social factors.“, line 435-439.

11. The conclusion needs a significant revision. It should be expanded and elaborative with the study's findings and future scope.

Once again, we agree and have changed the wording to: 

“A relatively high percentage of very old people had a negative impact on mental health from social distancing related to the COVID-19 pandemic. The cause is probably multifactorial, and this study showed that factors such as perceived loneliness and difficulty adhering to social distancing measures are important. Still, also resilience factors such as high morale seemed to be protective. These factors are important if future pandemics' negative impact on mental health for older people should be prevented.”, line 506-511.

Reviewer #2: 

1. Study fail to capture previous research related to mental health suffer with contemporary discourses and related facts

Thank you for your opinion. We kindly disagree. We believe that this journal and this article benefit from a short introduction and that we have presented the necessary background material from previous research. If there are specific references we have missed, please point them out or in a specific direction of what is missing. 

2. Association between variable and data size in row and column is not statistically corrected. It is applicable for table 1, table 2, and table 3. Binary logistic method also needs to be crosschecked. Finding in table 4 is doubtful. please check again and see the value of odd ratio and corresponding interval value.

We thank you for this, but we don’t understand the problem. The tables are not row-total, they are column-total, for instance in Table 1, where the percentage of females in the two groups does not add up to 100% (64.0% and 49.7%), but this is because there were, for instance, 64.0% female in the group with a negative impact on mental health (47 participants of 75 participants is 64.0%). 

3. Manuscript has gone through various incorrect sentences and wrong sentence frame; Manuscript needs a lot of rigorous grammatical check.

The manuscript was sent to a well-known and very competent proofreading company (San Francisco Edit). We have read the manuscript once again and made a few minor corrections. 

4. Abstract: The paper has a lot of problems with grammar that start right at the beginning. See “Of 394 eligible participants, 257 (65.2%) agreed to participate. Of these, 250 individuals reported mental health impact from COVID-19 and constituted the final sample”.

We don’t understand why there is concerns about this sentence. This sentence was approved by the proofreading firm. We have shortened it slightly due to wordcount limit after other changes in abstract. 

5. Abstract: In result section under abstract, it is not good to write detailed finding of odds ratio. Please revise the text with better sentence structure and clarity and make it concise.

Thank you for your opinion, but we kindly disagree. We believe the abstract should include a few details of the main finding, including the odds ratio to support argument in the conclusion.

6. Introduction: The current text exhibits a deficiency in connectivity and coherence both within individual sections and across different sections. There is a need to establish a connection between the two units. 

Thank you for your opinion. We kindly disagree. We believe there is coherence and connectivity between different sections in the manuscript.

7. The authors should incorporate relevant scholarly sources from previous studies. The introduction fails to effectively convey the essence of the title. The topic of mental health is noticeably lacking in the section on being home alone. The manuscript, to some extent, did not successfully convey the essence of the concept of being home alone. The precise definition of study is not provided. 

We agree that “Home alone” is a misleading part of the title and has now been removed, line 3. Further, introductions can be written in different ways depending on the journal, some very long and some short. We believe that this journal and this article benefit from a short introduction and that we have presented the necessary background material from previous research.

8. Data and Methodology: 250 people filled out the survey (line 103), but the numbers in Table 1, Table 2, and Table 3 do not match up in the rows and columns where they should. If the row data is 247, how is it split between columns 75 and 175? And it is also applicable for other numbers in row.

In every study, even the best researchers fail to capture all data, and there will be missing data. We have been transparent with this and added the numbers of individuals where data was missing. For instance, in Table 1, years of schooling was only 247 participants out of 250, and there were 74 out of 75 with data among those with negative impact, and 173 out of 175 with data. We did calculations with the data we had. If the reader wants to know the exact numbers, it could be calculated from the percentage. We chose to do this since including every number would make these table very hard to read. In our experience, this is how this matter is most often handled.

9. The subjective response is grouped together under the health-related factor. Does the author have any studies that show how to combine subjective responses with five groups into two? 

We believe it is common practice to handle different scales with multiple answering alternatives to make it easier to use in statistics, therefore we did not provide references to grouping of answer alternatives to each scale used in this manuscript. How the five answer alternatives are combined into two might differ on the purpose of the calculations and the distribution of answers. There will be a loss of precision, but it seems reasonable for this type of manuscript. 

10. Similarly, “Self-rated health compared to one year ago was based on the SF-36 item “compared to one year ago, how would you rate your health in general now” and responses are 0 = secure, 1 = insecure/hard to tell. Do author has any reference to club the category (insecure and hard to tell). 

Please refer to previous answer, number 9. We have provided a justification for grouping “hard to tell” with those who felt insecure. We thank you for pointing our attention to this, since we had an illogical reasoning. It has now been changed to: 

“The justification that individuals who stated that it was hard to tell whether they felt secure or not could not, likely were insecure.”, line 184.

11. Again, in reference to Geriatric Depression Scale, Philadelphia Geriatric Center Morale Scale Mini-Mental State Examination, how author decided the limit of score for particular case or event. Author does not provide any depth information related to score construction and related questions. 

The cut off for the Geriatric Depression Scale is 5 or more points, which is stated in line 153. We believed we didn’t have to restate the reference 23 – 24.

The PGCM scale was constructed by Lawton in 1972, and in the year 2003, he provided instructions on how to group the scores (in line 160-161) in the following reference already was provided:

Reference 26: Lawton MP. Lawton's PGC Morale Scale [Internet]. Polisher Research Institute Abramson Center for Jewish Life (formerly the Philadelphia Geriatric Center); 2003 [cited 2022-05-30]. Available from: https://abramsonseniorcare.org/media/1198/lawtons-pgc-moral-scale.pdf.

12. Also, the variable under "social factor" is not enough to explain why the categories of responses might be merged.

Please see our response to question number 9

13. It is also good to capture only relevant scale or score instead of multiple scale or score, and before that detailed and proper justification need to be address.

We partially agree and partially disagree with this statement. Too many variables might make it difficult to explain to the reader, and too few will not illustrate the whole picture. We believe the scales in this manuscript are not too many. 

14. Results: Table 1 consist of association of sample data i.e., 250 and they have answered the related question. This sample is again classified in Yes (N = 75) and No (N = 175). But when it associated with educational status (years of schooling) * (N = 247), Educational level (high)* (N=247), Place of residence (number of inhabitants) * (N = 239), and Residential status (residing alone) * (N = 244), these sample is not matched with column sample. It is again applicable for table 2 and table 3. It requires further analysis of data. 

We apologize, but we don’t understand what the problem is. Please refer to our answers, numbers 2 and 8.

15. In table 4 logistic regression was perform for various Sociodemographic factors, Health-related factors, and social factors. It seems that data analysis does not perform well. Odd ratio with lower and upper limit explains different story. I recommend for further analysis. 

We apologize once again we don’t understand what the problem is. Odds ratio presented are between upper and lower limit, and we believe they tell the same story. 

16. The author again highlighted various score and scale for mental health, but manuscript does not consider such information and calculating method of that score. 

Again, we do not understand the concern. 

17. GDS-15 score and MMSE score do not capture in multivariate analysis. Author should provide an analysis or information-why they are excluded from the analysis how it can be identified without any findings (Any variable with a univariate p-value of <0.15 was selected as a candidate for the multivariable analysis. LINE:213)

The MMSE was not included in the final logistic regression model since its p-value, which was 0.90, was above the limit of 0.15 that was necessary to be included. 

As mentioned in the statistics section, for statistical reasons we could not include all candidate variables but had to limit the variables added in the final model. Rule of thumb is that you can use one variable per 10 participants with the investigated outcome and since we had 75 participants with negative impact on mental health, we could use 7 or 8 variables in our regression model, which is stated in the manuscript. From previous knowledge we know that GDS-15 (i.e depression) correlate strongly with perceived loneliness, and there were fewer who had depressive symptoms than had perceived loneliness, we therefore chose only to use perceived loneliness in the final regression model.

18. Majorly all the tables have numerous findings, but author should interpret it under different section. It is missing in manuscript.

We are of the opinion that important results should be interpreted in the discussion section, but it would be tedious for the reader if every result was discussed. Please also see our answer to concern number 25.

19. Why chi2 test and FE fisher test used together? it is good to relay on specific method for all the variables.

Both tests are used to analyze dichotomized variables. “Chi-square Test for Independence” is the most commonly used, but there are assumptions that need to be fulfilled. One such is that variables should not be skewed, or at least 80% of cells with expected frequencies of 5 or more. If this assumption is violated, then most scientists use the “Fisher's Exact test” (in 2 by 2 tables). This is common practice. 

20. Discussion: In this section author used “very old”. Is it previously described in manuscript anywhere. Please refer it. 

We don’t understand this concern. Very old is used in the title: “Very old people”. In the first line in the Discussion section, we wrote “very old participants”, and under the design section, we explained how old our sample is (>80 years). 

21. Is 30 percent or any number is directly associated with table 1. 

We apologize but we don’t understand the concern. We found that 75 individuals experienced negative impact from COVID-19 out of 250 and calculated 75/250=0.30 i.e. 30 percent

22. line 309 to 313 is not evidence-based statement. Kindly refer it with previous research. 

We apologize, but we don’t understand the problem. We just state that having difficulty adhering to recommendations from authorities, the problem might have several different reasons, both cognitive and social seems reasonable. We added this statement to help the reader follow our argument through the long paragraph. Since we don’t know the true reasons for the problem why the participants experience problem to adhere to the guidelines we speculate over the reasons. We feel it is impossible and inappropriate in a manuscript like this to have a comprehensive debate on all possible reasons.

23. Please explain line 325 with various associated factor comes under GDS-15. The line “This could partly be explained by differences in data collection, measuring scales, and age of participants and may limit the comparability of study results. However, amongst respondents reporting a negative impact on mental health by the 330 COVID-19 pandemics,” could be the part of limitation of study.

We apologize, but we don’t understand the concern. We noticed our number of depressions were lower than in other research, but among those who had a negative impact on mental health from COVID-19 the number were similar. We can only speculate on the differences. Since GDS-15 is not included in the final regression model, we did not comment on this in strength and limitations.

24. Logistic regression model seems incorrectly estimated please check again. 

We apologize once again, but we don’t understand the concern. We have double-checked all the calculations. 

25. Even in discussion section odds value should be the part of discussion but it is missed majorly. 

We have been taught not to present data in discussion section, or at least keep it to a minimum. Therefore, we did not present odds values and they were already presented in the Results section.

We chose to discuss the variables with the highest Odds Ratio: perceived loneliness, difficulties in adhering to social distancing, the protective effect of high morale, and most surprisingly, that high educational level was significant. We also chose to discuss some variables not included in the model, such as GDS-15, sex, and residential status. We chose not to discuss all variables (like perceived negative subjective impact on the social situation) because long manuscripts are more difficult to get published. 

26. Please refer standard form of odds ratio, p value, lower limit and upper limit during framing the statement.

We believe we have used standard form used by the paper when presenting our results. 

27. Overall, this manuscript needs a lot of regress work under following section-

• Previous research and key definition of objective term

• Grammatical error and sentence framing

• Data cleaning and required variable.

• Data analysis and interpretation of findings

• Discussion and logical statement.

Once again, we thank you for your opinion, but we kindly disagree.

---

## [Decision Letter · Decision Letter 1]

10 Oct 2023

PONE-D-22-23481R1Association between the COVID-19 pandemic and mental health in very old peoplePLOS ONE

Dear Dr. Niklasson,

Thank you for submitting your manuscript to PLOS ONE. After careful consideration, we feel that it has merit but does not fully meet PLOS ONE’s publication criteria as it currently stands. Therefore, we invite you to submit a revised version of the manuscript that addresses the points raised during the review process.

Please see the comments made by Reviewer 2 and try to address them as much as possible.

We look forward to receiving your revised manuscript.

Kind regards,

Alok Ranjan

Academic Editor

PLOS ONE

Reviewers' comments:

Reviewer's Responses to Questions

**Comments to the Author**

1. If the authors have adequately addressed your comments raised in a previous round of review and you feel that this manuscript is now acceptable for publication, you may indicate that here to bypass the “Comments to the Author” section, enter your conflict of interest statement in the “Confidential to Editor” section, and submit your "Accept" recommendation.

Reviewer #1: (No Response)

Reviewer #2: (No Response)

2. Is the manuscript technically sound, and do the data support the conclusions?

Reviewer #1: Yes

Reviewer #2: No

3. Has the statistical analysis been performed appropriately and rigorously? 

Reviewer #1: Yes

Reviewer #2: No

4. Have the authors made all data underlying the findings in their manuscript fully available?

Reviewer #1: Yes

Reviewer #2: Yes

5. Is the manuscript presented in an intelligible fashion and written in standard English?

Reviewer #1: Yes

Reviewer #2: Yes

6. Review Comments to the Author

Reviewer #1: The authors of this article have provided clarification on many of the issues I brought up in my earlier review. If authors could address these :-

1.Topic : The authors could be more specific in the topic by stating the nation name, as the study is primarily focused on the context of Sweden.

2. Result : In order to analyse the statistical findings, the commentary of the tables in the result section needs to be expanded, especially section 3.4 - social factors.

Reviewer #2: I am grateful to the author for the response. Despite the fact that these responses cannot provide any valid evidence to support the raised concern. In the majority of cases, the author neither admits the validity nor provides supporting text or evidence. In descriptive statistical analyses where the author fails to identify missing responses, data cleaning is a major concern. I am again emphasizing an important review point (In previous Review) where the author does not cite any evidence-

Point 8- Data and methodology: cleaning of data and study sample.

Point 9, Point 10, and Point 12- Classification of categories under Subjective response and various other grouping categories--no prior research provides.

I-ADL and P-ADL: The author does not provide specific details for these categories.

Point-15: OR, LL, and UL estimates or highly questionable. [(sample mean) - (constant) x (SEM)] to [(sample mean) + (constant) x (SEM)] could be used to determine the Confidence Interval. Therefore, it may be closer to the mean value of CI.

Even in this investigation, the author provides no supporting evidence for the findings.

Even under table 4, section 3.5 Predictors of deteriorating mental health is not adequately discussed in detail.

Overall, the provided manuscript ignores the main correction that was previously communicated to the author. I believe that the manuscript is not suitable for my acceptance.

7. PLOS authors have the option to publish the peer review history of their article (what does this mean?). If published, this will include your full peer review and any attached files.

Reviewer #1: No

Reviewer #2: No

---

## [Author Response · Author response to Decision Letter 1]

28 Nov 2023

The second response to Reviewers

We thank once again the reviewers for their time and effort in reviewing our manuscript! Please find our response below.

Reviewer #1

1.Topic: The authors could be more specific in the topic by stating the nation name, as the study is primarily focused on the context of Sweden.

RESPONSE: We are not sure what “topic” refers to, but we have changed the title and aim so that it now states “in Sweden”. We hope this settles the concern.

2. Result: In order to analyze the statistical findings, the commentary of the tables in the result section needs to be expanded, especially section 3.4 - social factors.

RESPONSE: Thank you for pointing our attention to this aspect. We have made changes to section 3.4 in the results and also in parts of the discussion where we added a paragraph, particularly for discussing these findings. 

Reviewer #2: 

Reviewer #2 ask us to improve the answer to five previous concerns and two new ones, and we are happy to try to make a better answer! 

1. Previous point 8 Data and Methodology: 250 people filled out the survey (line 103), but the numbers in Table 1, Table 2, and Table 3 do not match up in the rows and columns where they should. If the row data is 247, how is it split between columns 75 and 175? And it is also applicable for other numbers in row.

Data and methodology: cleaning of data and study sample

RESPONSE: We are still not sure if we have understood your concern correctly, but we have now added information on the denominator in Tables 1, 2, and 3. 

For instance: Residential status (residing alone), among those with negative impact on mental health by COVID-19 there were 38 residing alone and 75 were able to answer, and among those without negative impact 79 residing alone of 169. That means that of (38+79=) 117 were residing alone and (75+169=) 244 who were able to answer the question. We now have presented all the values and the reader can make Chi-square calculations if desired to check our numbers. 

2. Previous points 9, 10, and 12 Does the author have any studies that show how to combine subjective responses with five groups into two? Do author has any reference to club the category Not enough to explain why the categories of responses might be merged.

RESPONSE: The use of dichotomization has been discussed over many years (1-3). It can be used to simplify data analyses and interpretation and to be able to compare. There are drawbacks such as loss of information, the cut-offs might not be obvious, different cut-offs can lead to different conclusions, decreased precision, etc. We have added to the manuscript a reason for using dichotomization in the method section and in the limitations discussed drawback of such procedure.

There are manuscripts that for instance combine five answer alternatives to two, evaluation of one’s health (4, look in Table 1).

1. Ben-Shakhar G. A further study of the dichotomization theory in detection of information. Psychophysiology. 1977 Jul;14(4):408-3. PMID: 882621.

2. Altman DG, Royston P. The cost of dichotomising continuous variables. BMJ. 2006 May 6;332(7549):1080. PMID: 16675816; 

3. Fedorov V, Mannino F, Zhang R. Consequences of dichotomization. Pharm Stat. 2009 Jan-Mar;8(1):50-61. PMID: 18389492.

4. Tak E, Staats P, Van Hespen A, Hopman-Rock M. The effects of an exercise program for older adults with osteoarthritis of the hip. J Rheumatol. 2005 Jun;32(6):1106-13. PMID: 15940775.

3. Previous point 15 OR, LL, and UL estimates or highly questionable. [(sample mean) - (constant) x (SEM)] to [(sample mean) + (constant) x (SEM)] could be used to determine the Confidence Interval. Therefore, it may be closer to the mean value of CI.

RESPONSE: This was a difficult concern for the authors, so we asked for help from an experienced statistician who gave this answer to provide in this response to Reviewer #2:

”As for the comment on how confidence intervals for odds ratios should be calculated, we agree that the suggested expression is correct for coefficients in a linear model. However, since we use odds ratios as is customary in logistic regression, this expression is not correct.”

We therefore conclude that the output generated from SPSS statistical software is accurate. 

4. New point 1 I-ADL and P-ADL: The author does not provide specific details for these categories.

RESPONSE: We thank the reviewer for this comment and agree that the reader could benefit from knowing the background for I-ADL and P-ADL. The following two sentences were included:

“To be deemed as independent in I-ADL the participants had to be independent in all the following activities: cleaning, shopping, transport, and cooking. To be deemed independent in P-ADL, the participants had to be independent in all the following activities: bathing, dressing, toileting, transfer, continence, and eating.”

5. New point 2 Even under table 4, section 3.5 Predictors of deteriorating mental health is not adequately discussed in detail.

RESPONSE: Thank you for pointing our attention to this aspect. We have made changes to section 3.4 in the results and also in parts of the discussion where we added a paragraph, particularly for discussing these findings.

---

## [Decision Letter · Decision Letter 2]

6 Feb 2024

Association between the COVID-19 pandemic and mental health in very old people in Sweden

PONE-D-22-23481R2

Dear Dr. Niklasson,

We’re pleased to inform you that your manuscript has been judged scientifically suitable for publication and will be formally accepted for publication once it meets all outstanding technical requirements.

Kind regards,

Mario Ulises Pérez-Zepeda, M.D., Ph.D.

Academic Editor

PLOS ONE

Additional Editor Comments (optional):

Reviewers' comments:

Reviewer's Responses to Questions

**Comments to the Author**

1. If the authors have adequately addressed your comments raised in a previous round of review and you feel that this manuscript is now acceptable for publication, you may indicate that here to bypass the “Comments to the Author” section, enter your conflict of interest statement in the “Confidential to Editor” section, and submit your "Accept" recommendation.

Reviewer #1: All comments have been addressed

Reviewer #2: All comments have been addressed

2. Is the manuscript technically sound, and do the data support the conclusions?

Reviewer #1: Yes

Reviewer #2: Yes

3. Has the statistical analysis been performed appropriately and rigorously? 

Reviewer #1: Yes

Reviewer #2: Yes

4. Have the authors made all data underlying the findings in their manuscript fully available?

Reviewer #1: Yes

Reviewer #2: Yes

5. Is the manuscript presented in an intelligible fashion and written in standard English?

Reviewer #1: Yes

Reviewer #2: Yes

6. Review Comments to the Author

Reviewer #1: This paper is significant as it deals with ageing population and their mental state during pandemic. This will helps to take specific measures for older population in the future.

Reviewer #2: Thanks for incorporating the comments and improve the manuscript. I also apricate your effort to provide citations with your responses.

7. PLOS authors have the option to publish the peer review history of their article (what does this mean?). If published, this will include your full peer review and any attached files.

Reviewer #1: No

Reviewer #2: No

---

## [Editor Report · Acceptance letter]

22 Mar 2024

PONE-D-22-23481R2 

PLOS ONE

Dear Dr. Niklasson, 

I'm pleased to inform you that your manuscript has been deemed suitable for publication in PLOS ONE. Congratulations! Your manuscript is now being handed over to our production team.

Kind regards, 

on behalf of

Dr. Mario Ulises Pérez-Zepeda 

Academic Editor

PLOS ONE